

# Antibiotic exposure enriches streptococci carrying resistance genes in periodontitis plaque biofilms

Qian Zhang[1], Min Zhen[2], Xiaochen Wang[3], FengXiang Zhao[4], Yang Dong[5], Xiaoya Wang[4], Shengtao Gao[4], Jinfeng Wang[4], Wenyu Shi[5] and Yifei Zhang[6]

[1] Central Laboratory, Peking University School and Hospital of Stomatology & National Center for Stomatology & National Clinical Research Center for Oral Diseases & National Engineering Research Center of Oral Biomaterials and Digital Medical Devices, Beijing, China
[2] Department of Periodontology, Peking University School and Hospital of Stomatology & National Center for Stomatology & National Clinical Research Center for Oral Diseases & National Engineering Research Center of Oral Biomaterials and Digital Medical Devices, Beijing, China
[3] CAS Key Laboratory of Animal Ecology and Conservation Biology, Institute of Zoology, Chinese Academy of Sciences, Beijing, China
[4] College of Food Science & Nutritional Engineering, China Agricultural University, Beijing, China
[5] State Key Laboratory of Animal Biotech Breeding, College of Biological Sciences, China Agricultural University, Beijing, China
[6] Department of Dental Materials, Peking University School and Hospital of Stomatology & National Center for Stomatology & National Clinical Research Center for Oral Diseases & National Engineering Research Center of Oral Biomaterials and Digital Medical Devices, Beijing, China

Corresponding authors
Wenyu Shi, shiwy@cau.edu.cn
Yifei Zhang, wingsflying@bjmu.edu.cn

## ABSTRACT

**Background:** Periodontitis is not always satisfactorily treated with conventional scaling and root planing, and adjunctive use of antibiotics is required in clinical practice. Therefore, it is important for clinicians to understand the diversity and the antibiotic resistance of subgingival microbiota when exposed to different antibiotics.

**Materials and Methods:** In this study, subgingival plaques were collected from 10 periodontitis patients and 11 periodontally healthy volunteers, and their microbiota response to selective pressure of four antibiotics (amoxicillin, metronidazole, clindamycin, and tetracycline) were evaluated through 16S rRNA gene amplicon and metagenomic sequencing analysis. Additionally, sensitive and resistant strains were isolated and cultured *in vitro* for resistance evaluation.

**Results:** Cultivation of subgingival microbiota revealed the oral microbiota from periodontitis patients were more resistant to antibiotics than that of healthy. Significant differences were also observed for the microbial community between with and without antibiotics (especially amoxicillin and tetracycline) treated in periodontitis group.

**Conclusion:** Overall, after the two antibiotics (amoxicillin and tetracycline) exposed, the oral subgingival microbiota in periodontitis patients exhibited different diversity and composition. *Streptococcus* may account for oral biofilm-specific antibiotic resistance in periodontitis. This provides information for personalized treatment of periodontitis.

## INTRODUCTION

Periodontitis, a chronic infectious disease, can result in gingival tissue inflammation, alveolar bone loss and eventually tooth loss. The key etiology was pathogenic microorganisms which can form biofilms (*Marsh, Moter & Devine, 2011*). According to the latest estimates in 2019, there were about 1.1 billion prevalent cases of severe periodontitis worldwide (*Chen et al., 2021*), reflecting a great economic burden of dental care. If not treated properly, periodontitis would impact not only oral health, but also systemic health outcomes, which can lead to a great economic and mental burden of patients (*Orlandi et al., 2021*). Though most patients can obtain satisfactory effect through scaling and root planing (SRP) (*Cobb, 2002*), it often be suboptimal due to the inability to access deep pockets, surface irregularities and furcation areas (*Jepsen et al., 2011*). Thus adjunctive antibiotics have been used locally or systemically to SRP for improving the clinical outcomes, including amoxicillin, metronidazole (*Ramos et al., 2022*; *Teughels et al., 2020*), clindamycin (*Luchian et al., 2021*) and tetracycline (*Szulc, Zakrzewska & Zborowski, 2018*).

Although patients with severe periodontitis would benefit the most from the adjunct use of specific antibiotics (*Walters & Lai, 2015*), the occurrence of antibiotic resistance during treatment has been a concern. Many researchers have reported that subgingival bacterial isolates exhibiting resistance to one or more antibiotics that frequently be used in periodontal therapy, especially for potential periodontal pathogens (*Ardila, Bedoya-García & Arrubla-Escobar, 2023*), *e.g.*, *Fusobacterium* and *Prevotella* species that resistant to metronidazole (*Ardila, Granada & Guzmán, 2010*); *Prevotella* spp. that resistant to amoxillin (*Arredondo et al., 2020a*; *Rams, Degener & van Winkelhoff, 2014*), clindamycin (*Rams, Degener & van Winkelhoff, 2014*) and tetracycline (*Arredondo et al., 2020b*). Those species, which exhibited variable antibiotic susceptibility, would potentially compromise the efficacy of the antimicrobial therapy and increase the risk of a clinical treatment failure and dysbiosis. More seriously, periodontal bacteria may ectopically colonize in other body sites through hematogenous or enteral route, and have also been found in cardiovascular and digestive system diseases (*Kitamoto et al., 2020*). Above all, the S3 Level Clinical Practice Guideline (CPG) has been developed to treat the periodontitis based on the antibiotic stewardship (*Sanz et al., 2020*).

It has been widely known that different diversity of microbiota between periodontitis and healthy controls. So, we hypothesized that the antibiotic resistance of the subgingival micro-environment in periodontitis patients would be distinct when treated with different antibiotics. Above all, the information about antibiotic resistance in periodontitis-associated microbiota will provide a guideline for the antibiotic personalized treatment programs of periodontitis.

## MATERIALS AND METHODS

### Subjects

Patients with periodontitis and periodontally healthy subjects aged 18~65 years were recruited by a qualified periodontist from Peking University School and Hospital of

Stomatology from 2018~2019. This study was approved by the Ethics Committee of Peking University Hospital of Stomatology (PKUSSIRB-201837098), and consent was obtained from all participating patients in accordance with the approved ethics protocol.

The inclusion criteria of the periodontitis patients were: no systemic disease, clinical attachment loss detected in the interproximal surfaces of more than two non-adjacent teeth, with a probing pocket depth ≥6 mm (*Papapanou et al., 2018*). The inclusion criteria of the healthy subjects were: no systemic or oral disease, probing depth ≤3 mm, bleeding on probing in <10% of sites, no attachment loss, and antibiotics-free within 3 months (*Chapple et al., 2018*). The exclusion criteria were pregnancy, smoking, systemic disease or antimicrobial therapy during the past 3 months, or periodontal therapy within 1 year. Examinations were conducted by a specialized periodontist.

## Sample collection and cultivation procedures

Recruited volunteers were requested no food and water for 2 h before sampling. Subgingival plaque samples were collected from the bottom of the periodontal/gingival pocket using a sterile Gracey curette. For each periodontitis patient, plaque from four diseased sites in different quadrants with a probing pocket depth ≥6 mm and attachment loss >0 mm. For each periodontally healthy control, plaque was collected from four healthy, no probe-bleeding sites in different quadrants. The plaque (four sites) from one volunteer was pooled in individual tube containing 500 μL phosphate buffer solution (PBS, Hyclone, Logan, UT, USA), transferred to the lab with ice box and processed within 30 min after sampling. Brain Heart Infusion medium supplemented with haemin (5 mg/L) and vitamin K (1 mg/L) (BHI-HV) was used as culture broth. Antibiotics were purchased from Solarbio (Beijing, China). Samples were shaken for 30 s and diluted 1:100 in 3 mL BHI-HV with or without amoxicillin (2 μg/mL), metronidazole (16 μg/mL), clindamycin (4 μg/mL), and tetracycline (8 μg/mL). The concentration values for these antibiotics were defined based on previous periodontal microbiology studies (*Feres et al., 2002*; *van Winkelhoff et al., 2005*; *van Winkelhoff et al., 2000*). Since we found the MIC of metronidazole against *Fusobacterium nucleatum* ATCC 25586 was 16 μg/mL in another experiment, we choose 16 μg/mL as the breakpoint concentration value for metronidazole instead of 8 μg/mL that reported in the reference. All incubation was performed in an anaerobic chamber (5% $H_2$, 5% $CO_2$, and 90% $N_2$) at 37 °C for 48 h. For each sample, 500 μL cultures were frozen in 10% skim milk at −80 °C for further isolation experiments.

## 16S rRNA gene amplicon and metagenomic sequencing

Bacteria were harvested by centrifugation (8,000 rpm, 10 min), followed by DNA extraction using a QIAamp DNA Mini Kit (Qiagen, Hilden, Germany). The quality was identified by 1% agarose gel and Qubit Fluorometer (Invitrogen, Waltham, MA, USA). High-quality DNA ($OD_{260}/OD_{280}$ = 1.8–2.0) with concentration >20 ng/μL were used for 16S rRNA gene V3-V4 region amplification. Additional details in sequencing and data processing are provided in SI Materials and Methods. Considering the gnomic quality of samples and meanwhile the patients must be resistant to the two antibiotics, so we selected five periodontitis patients with 15 samples to conduct metagenomic shotgun sequencing (five samples from amoxicillin treatment, five samples from tetracycline treatment and five

samples from non-antibiotics treatment). The raw sequencing data used in this study have been deposited in the NCBI SRA database under PRJNA952804. Additional details in sequencing, metagenomic analysis and antibiotic resistance gene identification are provided in SI Materials and Methods.

### Isolation of tetracycline-sensitive and -resistant strains

To isolate tetracycline-sensitive strains, approximately 50 µL frozen stocks of bacteria cultured in medium without antibiotic were spirally spread on agar plates of BHI-HV supplemented with 5% sheep blood (BHI-HVB) and incubated at 37 °C for 4 days. Colonies of different morphologies and colors were picked, and each colony was spread on BHI-HVB agar without or with 8 µg/mL tetracycline and incubated as above. Strains that grew only on BHI-HV agar were considered tetracycline sensitive, while strains grew on BHI-HVB agar without or with 8 µg/mL tetracycline were considered tetracycline-resistant. For another tetracycline-resistant strains, 50 µL freezer stocks of bacteria cultured in tetracycline-containing medium were spirally spread on BHI-HVB agar and incubated at 37 °C for 4 days. Grown strains were all considered resistant. Individual colonies were cultured in BHI-HV medium, identified by 16S rRNA gene Sanger sequencing (27F, 1492R), and freezer stocks were prepared for storage.

### *In vitro* evaluation of resistance phenotypes and *tet*M fragments

An agar dilution method was used for antibiotic-susceptibility testing. Additional details are provided in SI Materials and Methods. Isolates with different tetracycline sensitivities were cultured at 37 °C overnight, and genomic DNA was extracted using a DNA extraction Kit (Qiagen, Hilden, Germany). Polymerase chain reaction (PCR) amplification was performed using the genomic DNA as the template to confirm the presence of the *tet*M fragment. The primer sequences are listed in Table S1.

### Quantification of gene expression

Bacteria were cultivated with or without tetracycline (64 µg/mL) to the logarithmic phase respectively, and were harvested by centrifugation (12,000 rpm, 1 min), resuspended in 1 mL TRIzol reagent, heated at 95 °C for 15 min, and the remaining steps were conducted according to the manufacturer's instructions. The quality was identified by Nanodrop 8000 and Qubit Fluorometer (Invitrogen, Waltham, MA, USA). The high-quality RNA ($OD_{260}$/$OD_{280}$ = 1.8–2.0 and concentration > 300 ng/µL) was reverse transcribed to cDNA using ReverTra Ace qPCR RT Master Mix (Toyobo, Osaka, Japan). Quantitative real-time PCR was performed using SYBR Green Master Mix (Foreverstar Biotech, Beijing, China) in an ABI 7500 system (Thermo Scientific, Waltham, MA, USA). The primer sequences are listed in Table S1. The reaction volume was 20 µL, and the conditions were initial denaturation for 10 min at 95 °C, followed by 40 cycles of 10 s at 95 °C and 1 min at 55 °C for fluorescence collection, with a final extension for 1 min at 72 °C. The *tet*M gene of bacteria without tetracycline treated was used for normalizing. All assays were conducted in duplicate.

## Sequencing data analysis

16S rRNA gene amplicon sequencing data were processed using QIIME v1.9.1 (*Caporaso et al., 2010*). Metagenomic sequencing data were assembled into contigs using MEGAHIT (*Li et al., 2015*), and annotated against comprehensive antibiotic resistance database (CARD) 3.0.029 (*Alcock et al., 2023*) using DIAMOND (*Buchfink, Xie & Huson, 2015*). Reads coverage was evaluated by using Bowtie v10.3.0 (*Langmead & Salzberg, 2012*) and SAMtools (*Li et al., 2009*).

All gene sequences from human microbiome project (HMP) database (https://www.hmpdacc.org/HMASM/#data) were downloaded and blasted against *tet*M gene to screen out the target sequences. A total of 305 metagenomic sequenced samples associated with periodontitis and healthy statuses and their corresponding clinical information was obtained to assess the distribution of *tet*M (Table S2). Additional details in data analysis are provided in SI Materials and Methods.

## Statistics

Alpha diversity and Bray-Curtis distances were calculated using the R package Vegan. Principal coordinate analysis (PCoA) was performed, with the significance was determined by the nonparametric Adonis test ($P < 0.05$). LEfSe was used to identify the enriched or depleted bacteria between groups. Linear discriminant analysis (LDA) score >2.0 were considered as a discriminatory signature between different antibiotic treatments. Significance comparisons between groups were conducted by t-test and the multiple comparison FDR control method (Benjamin-Hochberg).

# RESULTS

## The effect of antibiotics on oral microbiota in periodontitis

Ten patients with periodontitis and eleven periodontally healthy subjects aged 18~51 years were included. The gender (male/female) distribution was 1:1 and the age and gender were calculated with no significant difference between the two groups.

By cultivation, there were more drug-resistant samples from the periodontitis group (P-group) compared with the periodontally healthy group (H-group) after exposure to each of the four antibiotics. All P-group samples exhibited growth when exposed to different antibiotics, while all H-group samples were susceptible to amoxicillin, six were susceptible to tetracycline, two were susceptible to clindamycin, and one was susceptible to metronidazole. Totally, 85 samples (including 21 without antibiotic treatment) with bacterial growth were enrolled for 16S rRNA amplicon sequencing and strain isolation (Fig. 1A and Table S3).

As a result, in the P-group, the microbial diversity decreased significantly after exposure to amoxicillin, clindamycin, and tetracycline, but not metronidazole, compared with H-group ($p = 0.038$, $0.015$, $0.002$, and $0.148$, respectively, t-test). In contrast, no significant differences were observed in the H-group ($p = 0.584$, $0.112$, and $0.506$ for metronidazole, clindamycin and tetracycline, respectively) (Fig. 1B). We next measured the inter-group community dissimilarities pre- and post- antibiotic exposure based on PCoA and pairwise

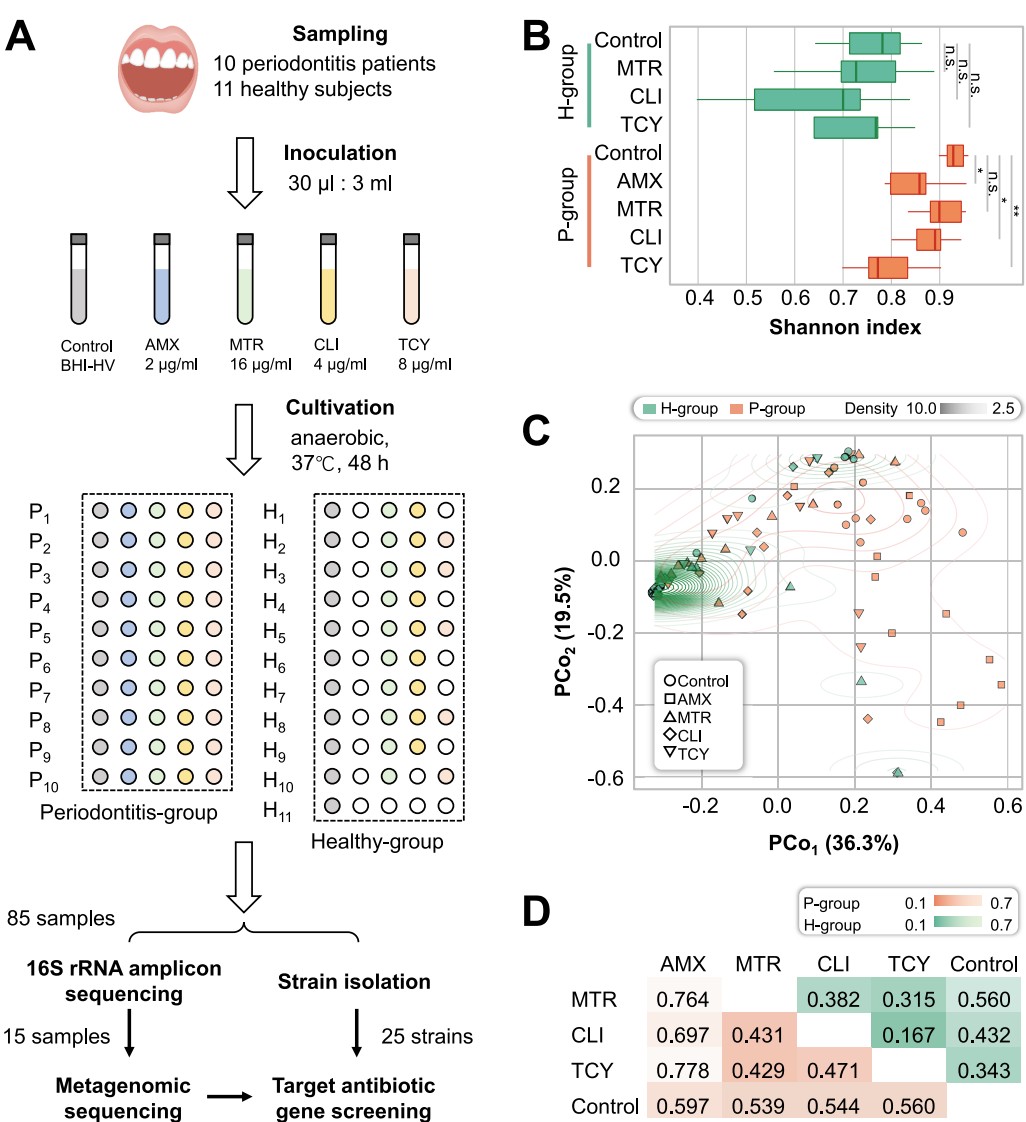

**Figure 1 Microbial diversity for periodontitis and healthy subjects treated with tetracycline, amoxicillin, metronidazole, and clindamycin.** (A) Study design. The five different color represented samples treated with four antibiotic and without antibiotic treatment. The white circles showed the bacteria were no growth in samples under different treatments. (B) Shannon diversity index for healthy and periodontitis groups treated with amoxicillin, metronidazole, clindamycin and tetracycline. n.s.: not significant; *$p < 0.05$; **$p < 0.01$. (C) The inter-group community dissimilarities by principal coordinate analysis (PCoA) using normalized OTU abundance among four treatments (amoxicillin, metronidazole, clindamycin and tetracycline) and control in healthy and periodontitis groups. The contour line showed the dispersion of samples in the groups. (D) The intra-group variation by Bray-Curtis distances in healthy and periodontitis groups. Bray-Curtis distances between paired samples in the same groups were determined using the R package ecodist. Statistical significance was determined by t-test and nonparametric Adonis test. H, healthy; P, periodontitis; AMX, amoxicillin; MTR, metronidazole; CLI, clindamycin; TCY, tetracycline. Control: samples from health-group and periodontitis group without antibiotic treatment.                     

Bray–Curtis distance. The results showed that the samples of H-group after exposure to antibiotics were closed to the control, while the samples of P-group were separated from their control (Figs. 1C and 1D).

## Periodontitis-specific antibiotic-resistant bacteria flourish after antibiotic exposure

In view of the different response of healthy and periodontitis-derived microbiota to antibiotic exposure, we then compared bacterial taxa between these two group with and without antibiotics. Firmicutes were the most abundant phylum regardless of antibiotic treatment or not, and Bacteroidetes and Fusobacteria, on the other hand, showed periodontitis- and antibiotic-specificity and were enriched only in the P-group (Fig. 2A).

There were remarkable inter-group variations in the bacterial genera resistant to each antibiotic based on LDA of the effect size. After exposure to amoxicillin, Prevotellaceae, Flavobacteriia, and Leptotrichiaceae of Bacteroidetes were enriched. Firmicutes, Aerococcaceae, Carnobacteriaceae, Streptococcaceae, and Lactobacillales bacteria were related to tetracycline resistance, which indicated that there was heterogeneity in antibiotic resistance among phylum members (Fig. 2B). Then, the relative abundance of genera in groups with and without antibiotic treatment was calculated. The prevalence of *Veillonella*, *Prevotella*, *Granulicatella*, *Haemophilus* genus were abundant in all the antibiotic-exposed periodontitis groups. Among the healthy subjects, only the genera *Virungella*, *Granulopsis* and *Klebsiella* had high relative abundance after exposure to clindamycin and tetracycline, while the genera *Granulopsis*, *Actinomyces* and *Klebsiella* had high abundance after exposure to metronidazole (Fig. 2C). *Streptococcus* were abundant in both groups, regardless of antibiotic exposure or not, which was the main taxon of tetracycline resistance in most samples except P8 (Fig. S1). Based on the 16S rRNA sequencing results, the diversity and relative abundance of genus was significantly decreased after amoxicillin and tetracycline exposed, which indicated amoxicillin and tetracycline were the most sensitive antibiotic in clinical therapy.

## Antibiotic exposure drives heterogeneity of antibiotic resistance genes carried by periodontal bacteria

Then we further investigated the causes of heterogeneity in antibiotic resistance in the oral microbiome. A total of 15 samples that met the requirements for metagenomic sequencing (P1, P3, P7, P8, and P10 with or without amoxicillin- and tetracycline-exposure in P-group) were used to analyze the effects of amoxicillin- and tetracycline-exposure on antibiotic resistance genes (ARGs) generation to uncover the causes of resistance differences. We assembled metagenomic reads into $2.1 \times 10^6$ contigs under the length cutoff of 500 bp. After gene prediction and ARGs annotation, 741 contigs were annotated with 149 ARGs among all the five samples (Fig. 3). Then we calculated the abundance of ARGs belong to each gene based on reads mapping coverage. We found that compared with control, amoxicillin-exposed samples were virtually unchanged, and four genes in tetracycline-exposed samples were significantly enriched (*t*-test, adjusted $p < 0.1$) (Fig. 3). Two of them were tetracycline-resistance gene *tet*M (Table S4). Accordingly, we next focus on *tet*M mediates tetracycline resistance.

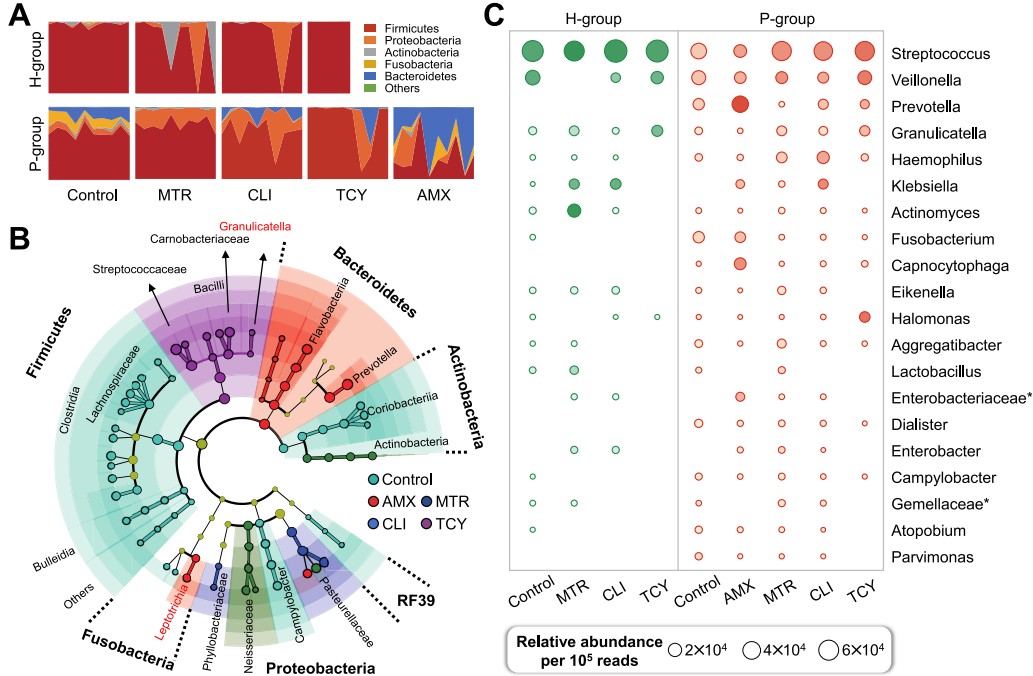

**Figure 2 Subgingival microbial composition of periodontitis patients and healthy subjects after antibiotic exposed.** (A) The relative abundances of bacteria at the phylum level in each sample from healthy and periodontitis groups with or without antibiotic treatments. (B) The phylogenetic distribution of microbial communities resistant to different antibiotic in periodontitis groups generated by LEfSe analysis. (C) The relative abundance of the top 20 bacterial genera in samples from healthy and periodontitis groups with or without antibiotic treatments. AMX, amoxicillin; MTR, metronidazole; CLI, clindamycin; TCY, tetracycline. Control: samples with no antibiotic added. H, healthy. P, periodontitis. * Represented the undefined genus belonged to family.

### *tet*M gene in *Streptococcus spp.* accounts for the tetracycline resistance

In order to measure the location of *tet*M-mediated tetracycline resistance in the body parts and which bacteria in humans are more likely to carry the gene, we retrieved *tet*M gene sequences from the HMP database and tracked their bacterial sources and the distribution of body parts. As a result, *tet*M can be detected in multiple body sites. The oral cavity was its main reservoir, especially for supragingival and tongue dorsum. Conversely, the detection rate was very low in stools. In addition, *tet*M was found to be widely distributed in various genera. For each body site, *tet*M carriage was restricted to a few genera. In oral cavity, *tet*M was mainly carried by *Streptococcus* (Fig. 4A).

Considering that *Streptococcus* genus harboring *tet*M might be resistant to tetracycline, enabling them to proliferate in the presence of tetracycline in oral microbiota, we isolated *Streptococcus* strains with different tetracycline-resistance phenotypic characteristics (Fig. 4B). Resistant strains were isolated mainly from tetracycline-exposure samples in P-group (Table S5). Resistant and sensitive strains of the same species co-existed in several samples (*e.g.*, P1S3 *vs*. P1m8; P7S2 *vs*. P7m8), indicating strain-level differences in resistance. As expected, all isolates lacking *tet*M were sensitive to tetracycline, while those

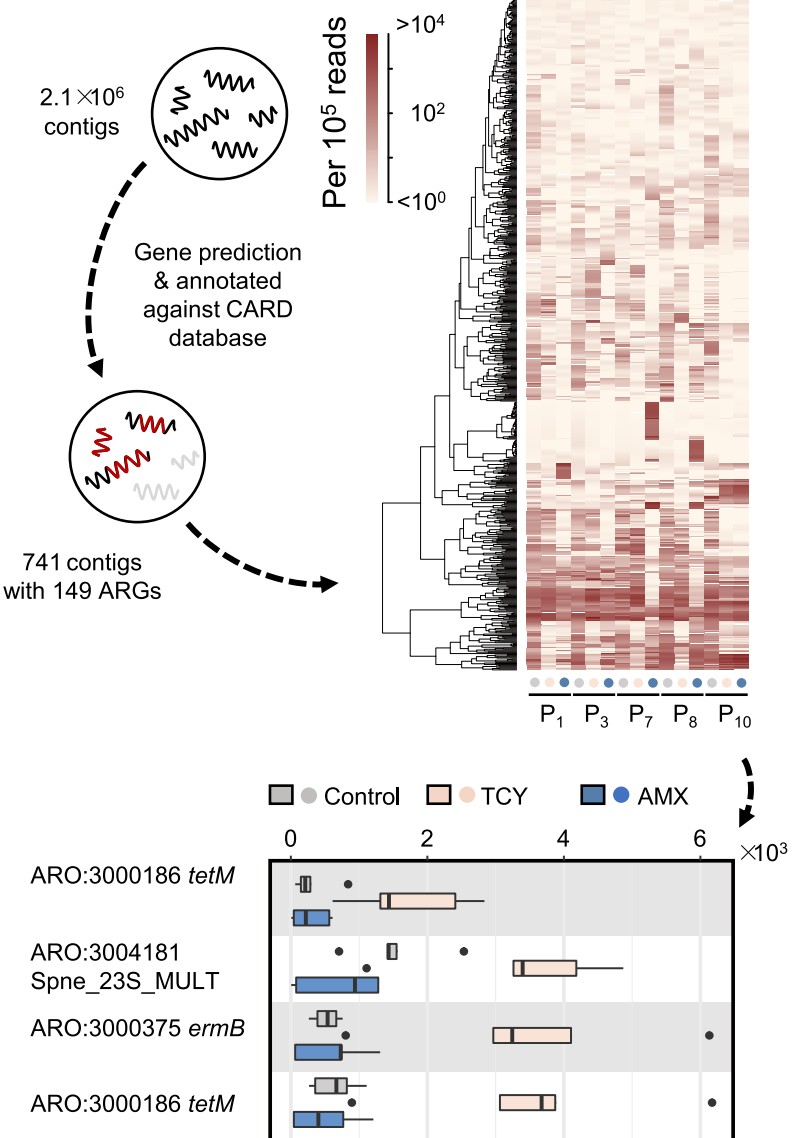

**Figure 3 Antibiotic-specific resistance gene annotated by metagenomic analysis.** The relative abundance of ARGs classified by resistance mechanism in five periodontitis samples treated with amoxicillin and tetracycline and the significant different contigs between the control and the samples with AMX and TCY treatments are shown. Contigs were blasted against antimicrobial resistance database CARD 3.0.029 to annotate ARGs using DIAMOND. P1, P3, P7, P8, P10 were samples which selected from periodontitis group. The control was samples from P1, P3, P7, P8, P10 with no antibiotic added. AMX: amoxicillin; TCY: tetracycline.

with *tet*M were resistant, except one *SI intermedius* strain (H11-2) isolated from sample H11, which contains *tet*M but was sensitive (Table S5 and Fig. S2).

To verify whether the *tet*M gene is also periodontitis-specific, 16 sequences were recruited from our isolates, which were *Streptococcus* genus, and found that *tet*M from periodontitis and healthy sources were priority clustered separately. While *tet*M from the same strain were not showed homologous (Fig. 5A). We also analyzed the relative

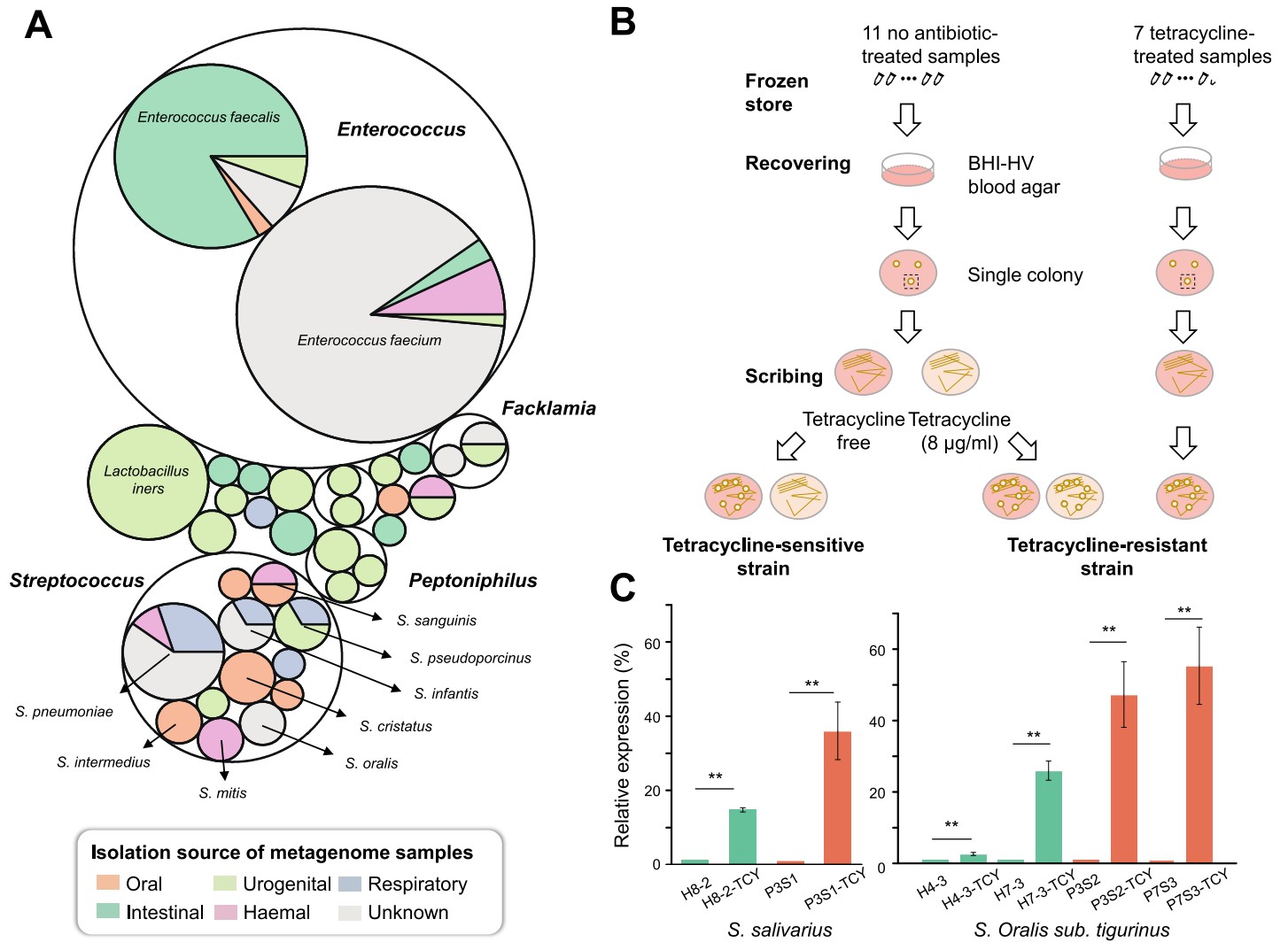

**Figure 4 Isolation and confirmation of antibiotic resistance gene in *Streptococcus spp.*** (A) The bacterial origin and the distribution of body sites of *tet*M gene based on HMP database. (B) The flow chart of TCY-sensitive and TCY-resistant strains isolation. (C) The relative expression of *tet*M in isolation strains before and after tetracycline added. *tet*M expression was normalized to the levels of 16S rRNA gene, then the fold of expression change was calculated. **Represented that tetracycline induced the expression of *tet*M highly significant difference ($p < 0.01$).

abundance of *tet*M in 305 metagenomic data of periodontally healthy subjects and periodontitis patients. Regardless of the sample type (saliva or plaque), the amount of *tet*M carried by healthy population was relatively stable, while the distribution of *tet*M among periodontitis patients varied greatly, with some patients carried more *tet*M genes (Fig. 5B).

To avoid inter-species discrepancy, we used isolates within the same species for comparison to test whether the periodontitis-derived *tet*M mediate stronger tetracycline-resistance. A total of 22 isolates belonging to five *Streptococcus* species were selected for MIC test. All tetracycline-resistant *SI sanginosus* isolates had the same MIC (32 μg/mL). While within the same species, *SI oralis sub. tigurinus*, *SI salivarius*, and *SI intermedius* isolates had varied MICs (ranged from 8 to 64 μg/mL) (Table S6), and the MICs of

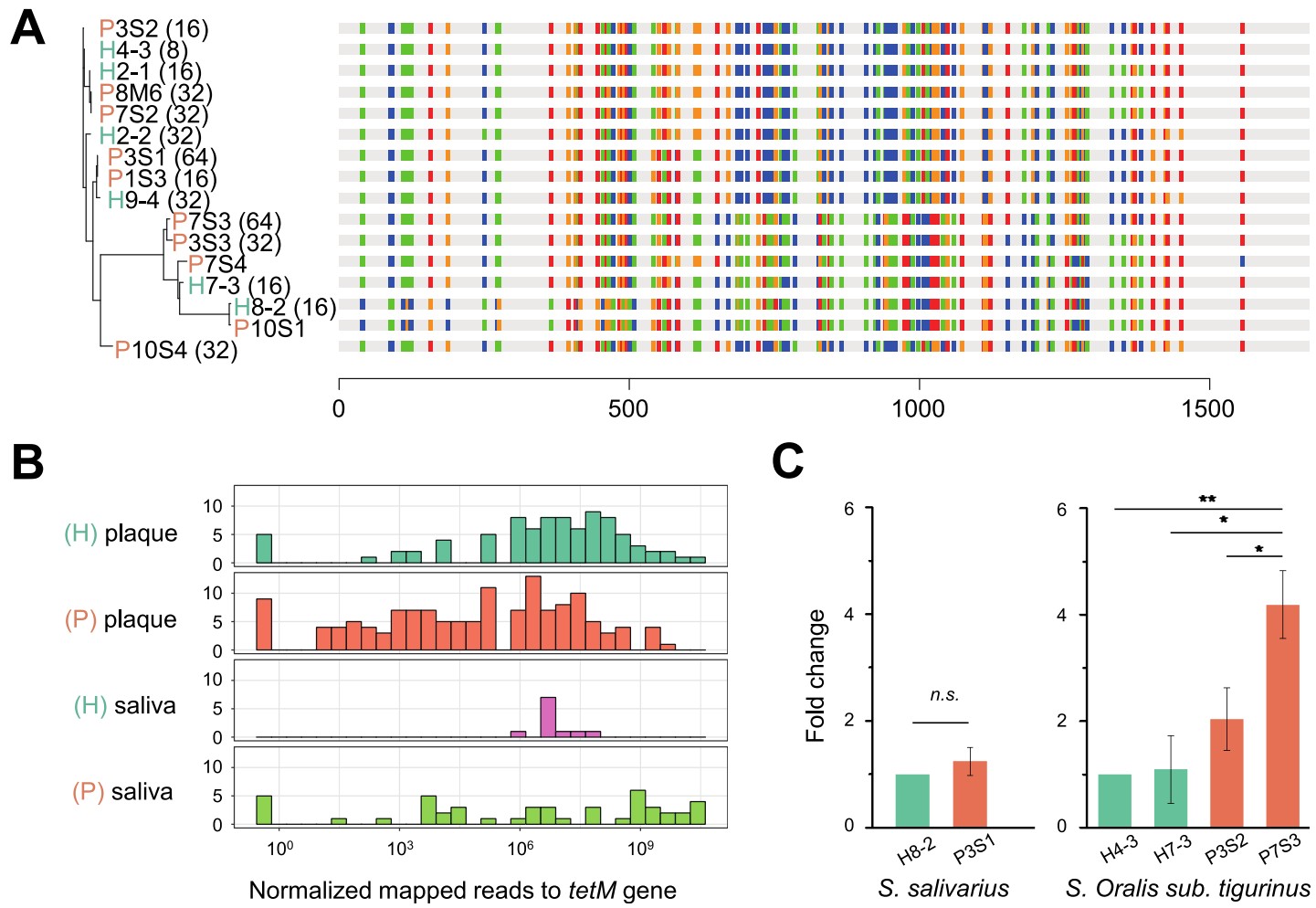

**Figure 5** **Phylogeny, distribution and expression of *tet*M gene.** (A) The phylogenetic tree analysis of *tet*M gene sequences from 16 isolated strains from *Streptococcus* genus with minimum inhibitory concentration (MIC) value. The unit of MIC was µg/mL. The four colors represented the A, T, C, G bases. (B) The distribution of *tet*M gene in saliva and plaque from healthy and periodontitis patients by screening metagenomic database (C) The fold change of *tet*M gene in the same isolated strains from periodontitis patients compared to form healthy volunteers under tetracycline treatment (64 µg/ml). H, healthy; P, periodontitis; * Represented that tetracycline induced the expression of *tet*M significantly ($p < 0.05$), ** Represented that tetracycline induced the expression of *tet*M highly significant difference ($p < 0.01$).

periodontitis-derived strains were higher or equal to that of healthy-derived strains. Finally, we evaluated the expression of *tet*M in strains with higher tetracycline exposure (64 µg/mL). The results showed that the expression of *tet*M at the transcriptional level in different strains was consistent with the resistant phenotypes of the strains, and strains derived from periodontitis expressed higher or equal to that of healthy-derived (Fig. 5C).

## DISCUSSION

The use of antibiotics in periodontal disease therapy is controversial because patients may harbor multiple periodontal pathogens with varying antibiotic susceptibilities, and the therapeutic benefit is compromised if the antibiotic does not reach the required concentration. Most oral bacteria live in biofilms, which increases their MICs 100- to
1,000-fold compared with planktonic bacteria (*Ceri et al., 1999*). Various efforts have clarified the complex interactions between microbes and dispersal of ARGs in biofilms, which indicated a co-selection phenomenon exists in the resistome of dental plaque microbiota (*Carr et al., 2020*; *Kang et al., 2021*). It was known that there was different subgingival microbiota in periodontitis patients and healthy subjects. Thus distinct microbiota will be observed in subgingival plaque from periodontitis patients and healthy subjects when antibiotic exposed. Here we conducted an *ex-vivo* experiment and found that antibiotic-resistant bacterial communities were widespread in subgingival plaque even without the history of antibiotic use in periodontitis patients.

The diversity of antibiotic-resistance determinants in oral biofilms reportedly differ according to periodontal status (*Arredondo et al., 2020b*; *Kim, Kim & Lee, 2011*). In this study, the difference was a result of significant structural differences in subgingival microbial communities between healthy individuals and periodontitis patients; some resistant members were detected only in the subgingival microbiota of patients with periodontitis (*e.g.*, Bacterioidetes that are resistant to clindamycin and amoxicillin). At the genus level, *Streptococcus* dominated the drug (metronidazole, clindamycin and tetracycline) resistance in healthy and periodontitis subjects, and mediated amoxicillin-resistance only in periodontitis patients, confirming its role as a multi-drug antibiotic-resistance reservoir. *Streptococcus*, as the pioneer species in the oral biofilm forming would influence the inter-species correlations which can change the response of multi-strain biofilm to antibiotics: the interactions within a community could improve its members' resistance to antibiotics, increase their antibiotic tolerance, change the expression of resistance genes, and allow individual member to express antibiotic defense to protect the whole community (*Bottery, Pitchford & Friman, 2021*; *Lee et al., 2014*; *Radlinski & Conlon, 2018*) or influencing the evolution of resistance (*Adamowicz et al., 2020*), and lead to the increase of minimal selective concentration when embedded in the community (*Klümper et al., 2019*).

According to our results, metronidazole is still the best choice adjunctive to severe periodontitis therapy since after metronidazole treatment the disease-derived microbial community was recovered as a healthy one. Most metronidazole-resistant bacteria are associated with periodontal health such as *Streptococcus*, *Actinomyces*, and other Gram-positive cocci and rods, and the resistance to metronidazole was intrinsic, limiting its spread and clinical impact. We highlighted the high proportion of amoxicillin-resistant Gram-negative anaerobic bacteria in disease-derived plaque samples, and supported clindamycin could be an ideal alternate to amoxicillin. While the potential side effects should also be considered when applied to the clinical practice. The pathogen *Porphyromonas* existed in low abundance in all untreated samples, and further reduced after antibiotic treatments (Table S7), indicating its susceptibility to antibiotics. Other studies have confirmed this (*Arredondo et al., 2020b*; *Veloo et al., 2012*).

To further elucidate resistance mechanisms, we targeted one particular ARGs (tetM), which plays a role in tetracycline resistance by comparing the relative abundance of ARGs in samples with and without antibiotic treatment. Further experiments confirmed that *Streptococcus* strains only carrying *tet*M were resistant to tetracycline. The expression of

resistance genes induced by low dose antibiotics have been supported by other studies (*Bowen et al., 2021*; *Jiang, Liu & Zhang, 2021*). As a mobile element, *tet*M is classified in high-risk rank of ARGs (*Zhang et al., 2021*), and oral biofilms may facilitate the horizontal transfer of ARGs (*Madsen et al., 2012*). The pollution of environment also act as a selective pressure to promote the horizontal transfer of ARGs. Combining with metagenomic analysis in human microbiome database, our study showed that *tet*M mainly exists in *Streptococcus* in oral cavity. However, the metagenomic analysis revealed that *tet*M can also be carried by other genera in other body sites, confirming the *in-vivo* transmission risk. Thus, there is a clear need for the development of guidelines to encourage rational antibiotic use in dentistry. Recently, a pharmacologic strategy termed "host-modulation therap"-that using non-antibiotic formulation of tetracycline was applied clinically in USA and Canada. This novel therapy for periodontal disease was based on the non-antibiotic, anti-collagenolytic properties of tetracycline (*Golub & Greenwald, 2011*; *Monk, Shalita & Siegel, 2011*). Besides periodontitis, a series of inflammatory/collagenolysis diseases, such as arthritis, diabetes, cardiovascular and lung diseases, can also benefit from the tetracycline-based host regulators (*Golub et al., 2016*). However, although these drugs have no antibiotic activity, it is still necessary to consider whether they have the ability to affect the expression and transfer of the tetracycline-resistant genes, since non-antibiotic compounds may also promote horizontal transfer of ARGs (*Alav & Buckner, 2024*).

In a polymicrobial environment, the antibiotic tolerance of a strain can be modulated by ecological interactions such as competition for nutrition, cross-feeding, and quorum sensing (*Adamowicz et al., 2018*; *de Vos et al., 2017*). In our study, the results showed that the tetracycline-resistance of *tet*M-carrying strains isolated from periodontitis sites was higher than that from healthy sources, which indicated that the variation of *tet*M is more closely related to the periodontal status than different species, while it is expected that the other *tet*-related genes may also be correlated to confer tetracycline-resistance (*Mullany, Allan & Warburton, 2012*).

However, the small sample size and lack of experiments using mutant strains prevented us from determining a direct relationship between Streptococci-borne *tet*M and tetracycline resistance. In addition, microbial interactions may influence drug susceptibility and whether resistance is exerted or enhanced through collaboration between microbes carrying and not carrying *tet*M genes in biofilms, the present study provides limited information in this regard. So the future investigation is needed, for example: longitudinal resistance dynamics studies, biofilm-associated resistance mechanisms studies, the expression level in different bacterial genera in periodontitis and inter-species interactions *in vivo* studies should be conducted to make the findings more clearly. Our study could be useful as a guideline for clinical therapeutic practice, not only for periodontitis, but also for the all dentistry.

## CONCLUSION

The study revealed the different antibiotic resistance in subgingival microbiota from periodontitis patients. *Streptococcus* in oral cavity accounts for periodontitis-specific tetracycline resistance. These findings highlight the antibiotic resistance profiles of

non-periodontal pathogens in periodontitis patients, which will provide information about tailored antibiotic approaches, and also facilitate personalized antibiotic treatment (microbial profiling, narrow-spectrum antibiotics or non-antibiotic antimicrobials) for severe periodontitis, and for practices in dentistry.

### Funding
This work was supported by the National Key R&D Program of China (2022YFA1304102), the National Natural Science Foundation of China (T2341010, 32070122, 32170659), and the 2115 Talent Development Program of China Agricultural University. The funders had no role in study design, data collection and analysis, decision to publish, or preparation of the manuscript.

### Grant Disclosures
The following grant information was disclosed by the authors:
National Key R&D Program of China: 2022YFA1304102.
National Natural Science Foundation of China: T2341010, 32070122, 32170659.
The 2115 Talent Development Program of China Agricultural University.

### Competing Interests
The authors declare that they have no competing interests.

### Author Contributions
- Qian Zhang conceived and designed the experiments, performed the experiments, authored or reviewed drafts of the article, and approved the final draft.
- Min Zhen performed the experiments, prepared figures and/or tables, and approved the final draft.
- Xiaochen Wang analyzed the data, prepared figures and/or tables, and approved the final draft.
- FengXiang Zhao analyzed the data, prepared figures and/or tables, and approved the final draft.
- Yang Dong analyzed the data, prepared figures and/or tables, and approved the final draft.
- Xiaoya Wang analyzed the data, prepared figures and/or tables, and approved the final draft.
- Shengtao Gao analyzed the data, prepared figures and/or tables, and approved the final draft.
- Jinfeng Wang analyzed the data, authored or reviewed drafts of the article, and approved the final draft.
- Wenyu Shi conceived and designed the experiments, analyzed the data, prepared figures and/or tables, and approved the final draft.
- Yifei Zhang conceived and designed the experiments, performed the experiments, authored or reviewed drafts of the article, and approved the final draft.

## Human Ethics

The following information was supplied relating to ethical approvals (*i.e.*, approving body and any reference numbers):

This study was approved by the Ethics Committee of Peking University Hospital of Stomatology (PKUSSIRB-201837098).

## Data Availability

The sequences are available at GenBank: PRJNA952804.

## Supplemental Information

Supplemental information for this article can be found online at http://dx.doi.org/10.7717/peerj.18835#supplemental-information.

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
