# Peer review of "Antibiotic exposure enriches streptococci carrying resistance genes in periodontitis plaque biofilms"

_PeerJ, doi:10.7717/peerj.18835_

## Round 0.1 · original submission · Major Revisions

The manuscript has been reviewed by two experts. While they noted that the study is of interest, both reviewers have provided detailed comments and recommend a number of changes to improve the clarity of the manuscript. Please address these concerns in a revised version of the manuscript. Reviewer #1 also suggests further statistical analysis of the data regarding differences between antibiotics.

Reviewer 1 ·

Basic reporting

1. Enhance Sentence Structure and Clarity
- Combine Related Sentences: Many sentences can be combined or streamlined for smoother readability.
- Avoid Repetition: Phrases like "Streptococcus dominated the drug resistance of healthy and periodontitis-derived microbial communities" could be simplified by removing redundancies. Try rephrasing to avoid repetitive information about Streptococcus and resistance patterns, making the text more concise.

2. Grammar and Syntax Adjustments
- Subject-Verb Agreement: In sentences like “The diversity and distribution of antibiotic-resistance determinants in oral biofilms reportedly differ according to periodontal status,” ensure subject-verb agreement throughout.
- Clarify Pronoun References: Pronouns such as "it" or "they" should clearly refer to specific nouns. For instance, in "Our study showed that tetM mainly exists in Streptococcus in oral cavity. However, it revealed that tetM can also be carried by other genera," clarify that "it" refers to "tetM" to prevent ambiguity.

3. Improve Flow and Transitions:
- Use Transitional Phrases: The discussion can benefit from more seamless transitions between ideas. For example, instead of “By comparing the relative abundance of ARGs...we targeted one particular ARGs (tetM),” use a phrase like “To further elucidate resistance mechanisms, we focused on one specific ARG, tetM…”
- Consistent Tense Usage: Ensure the use of past tense when describing the study’s procedures and findings. Switching between past and present tense can create confusion.

4. Refine Technical Language and Composition
- Technical Precision: Ensure technical terms are used precisely, e.g., replacing “does do not” with "does not" in "the therapeutic benefit is compromised if the antibiotic does not reach the required concentration."
- Use Active Voice Where Possible: Passive voice appears frequently in the discussion section. For example, instead of “A number of efforts have led to some understanding...,” consider rephrasing to “Various efforts have clarified the complex interactions…”

5. Polish the Conclusion
- Avoid Ambiguity and Increase Directness: In the conclusion, rephrase vague statements like "Our findings provide insights into the antibiotic susceptibility carried by non-periodontal pathogens on antibiotic therapy for periodontitis" for clarity. A more direct phrasing could be: "These findings highlight the antibiotic resistance profiles of both periodontal and non-periodontal pathogens in periodontitis patients, supporting tailored antibiotic approaches."
- Reinforce Key Points: Conclude with a clear, impactful statement summarizing the practical implications of the study’s findings.

6. Consistency in Terminology:
- Uniform Terminology for Microbial Names and Antibiotics: Ensure consistent use of microbial names, e.g., antibiotic names and resistance genes (e.g., tetM). Italicizing scientific names and ARG symbols, when appropriate, will add to the professional presentation of the text.

Experimental design

Title
1. Ensure the title clearly reflects the study’s primary focus and highlights the significant findings concisely.
2. Avoid potentially ambiguous terms like "periodontitis-specific antibiotic tolerance"; specify if it refers to tolerance unique to periodontitis patients or antibiotic resistance mechanisms linked to specific bacterial strains within biofilms.
3. Consider rephrasing for a more precise representation of the focus on Streptococcus and its role in antibiotic resistance within the context of dental biofilms in periodontitis.
4. Include the type of study
5. The title is very strong. dDue to the limitations of the study it should be presented more as a possibility. For example, Streptococcus in dental biofilms MAY.....

Abstract

1. Provide a more focused background by emphasizing the study's primary aim and reducing general statements on periodontitis and treatment challenges.
2. Streamline the "Materials and Methods" to concisely outline the study’s scope, such as patient sample size, antibiotic selection, and the microbiome analysis technique, ensuring clarity without excessive detail.
3. Ensure results section specifies key findings in a structured, clear manner, focusing on differences in antibiotic resistance between the periodontitis and healthy groups.
4. Clarify the conclusion to summarize the key implication of the findings regarding Streptococcus and antibiotic resistance, directly linking it to the potential impact on treatment strategies.
5. Verify that keywords are relevant to the core aspects of the study and correspond to standard MeSH terms for increased discoverability.

Introduction

1. Clarify the causal relationship in the first sentence by specifying that periodontitis is influenced by pathogenic microorganisms forming biofilms, contributing to chronic inflammation.
2. Provide additional context on why the prevalence and economic burden of periodontitis, as highlighted in recent estimates, underscore the importance of effective treatment.
3. Streamline the explanation on the limitations of scaling and root planing (SRP) by clearly linking specific challenges (e.g., deep pockets, surface irregularities) to the need for adjunctive antibiotic therapy.
4. Emphasize why antibiotic resistance poses a challenge in periodontitis management, including specific examples of resistant bacterial species, without overloading with excessive detail on individual species or antibiotic susceptibility patterns. There is also no reference to the recent developments in perio, for example, the EFP CPGs that clearly called for antibiotic stewardship (PMID: 32383274).
5. Rephrase complex sentences on systemic implications of untreated periodontitis for improved readability, explicitly linking periodontal bacteria migration to potential systemic health risks.
6. Strengthen the hypothesis by stating it in clear, concise language that directly addresses the anticipated differences in antibiotic resistance between periodontitis patients and healthy controls.
7. Conclude the introduction by highlighting the study’s significance, explaining how examining the diversity and antibiotic resistance in periodontitis-associated microbiota could lead to improved therapeutic strategies.
8. Line 53. Please revise planning (spelling).

Materials and Methods

1. Subjects:
- Expand on participant recruitment details to clarify any selection processes beyond the study site (e.g., random sampling or specific patient outreach methods).
- Specify the timeline for recruitment and any relevant demographic information about participants (e.g., gender distribution).
- Clearly define the inclusion and exclusion criteria for both periodontitis and healthy subjects, ensuring consistent terminology for clinical attachment loss and probing depths for precision.
- Line 87. “topical antimicrobial therapy during the past 3 months”. What about systemic therapy?

2. Sample Collection and Cultivation Procedures:
- Clarify the timing and handling of plaque samples to ensure sample consistency. For example, provide specific handling instructions, such as the number of samples processed at a time to minimize variability.
- Specify the exact protocol for PBS storage and transportation, if applicable, to reinforce sample stability before cultivation.
- Include more details on why certain antibiotics and concentrations were selected and provide context about the breakpoint concentrations used.

3. 16S rRNA Gene Amplicon and Metagenomic Sequencing:
- Provide a rationale for the use of both 16S rRNA gene sequencing and metagenomic sequencing, explaining how each method contributes to the analysis of microbiota diversity and resistance.
- Describe any quality control measures for the DNA extraction and amplification process to ensure reproducibility.
- Include a clear justification for selecting only five periodontitis patients for metagenomic sequencing, which would clarify sample representativeness and potential limitations.

4. Isolation of Tetracycline-Sensitive and -Resistant Strains:
- Clarify the methodology for isolating and identifying tetracycline-resistant strains, and explain why specific concentration levels were chosen.
- Specify controls used to validate sensitivity and resistance determinations for each strain, improving the rigor of resistance testing.

5. Quantification of Gene Expression:
- Define how gene expression was normalized, such as using housekeeping genes, to ensure consistency and comparability in qPCR.
- Include any steps taken to ensure RNA quality, such as quality checks after RNA extraction and before reverse transcription, and add more precise specifications for PCR conditions.

6. Sequencing Data Analysis:
- Expand on the data analysis tools and provide justification for choosing specific software (e.g., why QIIME for 16S rRNA data).
- Include a brief description of the sequence data processing pipeline, particularly for contig assembly and annotation.
- Clarify why particular databases (e.g., CARD and HMP) were used, explaining their relevance to antibiotic resistance studies and periodontitis research.

7. Statistics:
- Ensure that all statistical tests are thoroughly described and justified, explaining why each test (e.g., Adonis, t-test) was chosen and any assumptions they require.
- Provide more specific information on threshold values, such as exact P-values used for significance testing, to allow reproducibility.
- Include any corrections for multiple testing or additional statistical models used to interpret the microbiota data more robustly, particularly for diversity analyses.

Validity of the findings

Results

1. Further Analysis of Resistance Profiles Across Antibiotics: While the study provides insight into the resistance patterns in periodontitis versus healthy microbiota, it could benefit from additional statistical comparisons across all antibiotics tested. A more detailed analysis could clarify resistance trends across antibiotics within each group, strengthening the study’s findings on antibiotic effectiveness.

2. Broaden Interpretation of Microbial Diversity Findings: The reported reduction in microbial diversity within periodontitis samples following exposure to Amoxicillin, Clindamycin, and Tetracycline is significant. The authors should discuss the potential long-term implications of this diversity loss, as reduced diversity in the oral microbiome has been associated with various health consequences. Including this perspective may enhance the clinical relevance of the findings.

3. Consider Alternative Antimicrobial Approaches: Given the increase in resistant strains in the periodontitis group, it may be beneficial to discuss or recommend adjunctive or alternative therapies, such as narrow-spectrum antibiotics or non-antibiotic antimicrobials, as options to minimize resistance development. This could provide additional context for clinical strategies in treating periodontitis.

4. Detailed Characterization of Antibiotic Resistance Genes (ARGs): The enrichment of ARGs, specifically the tetM gene, is a key finding. Expanding the analysis to include expression levels across different bacterial genera in periodontitis samples would add depth to the understanding of resistance mechanisms in this context. Additionally, clarifying how ARG enrichment differs in periodontitis versus healthy samples could reinforce the implications for clinical antibiotic use.

5. Strengthen Discussion on the Clinical Implications of Resistance in Periodontitis-Associated Microbiota: Given the study's demonstration of more robust resistance in periodontitis samples, the authors could discuss how these findings should influence clinical antibiotic prescriptions for periodontitis. A detailed interpretation could help guide future antibiotic use recommendations specifically for dental practice.

6. Potential for Microbiota Profiling in Personalized Treatment: Based on the diverse response of microbial communities to antibiotic exposure, it may be worthwhile to suggest microbiota profiling as a tool for personalized treatment planning in periodontitis cases. This approach could support targeted therapy, reducing unnecessary exposure to broad-spectrum antibiotics and mitigating resistance risk.

Additional comments

Discussion
1. Clarify Antibiotic Selection Criteria: The discussion highlights the suitability of Metronidazole and Clindamycin for periodontitis but would benefit from a clearer explanation of why these antibiotics are preferred over others based on the study's findings. Expanding on the specific efficacy and resistance profiles observed in the study for each antibiotic could strengthen the rationale for their clinical use.

2. Expand on Biofilm-Related Resistance Mechanisms: The authors mention that biofilm formation increases the minimum inhibitory concentrations (MICs) of bacteria but do not fully explore how biofilm-specific mechanisms may contribute to the resistance patterns observed. Adding a more detailed explanation of how biofilms alter resistance phenotypes and discussing recent research on biofilm resistance mechanisms could improve the study’s contextual relevance.

3. Explore Broader Clinical Implications of ARGs in the Oral Microbiome: The identification of specific ARGs, like tetM, in periodontitis-associated bacteria is valuable. However, the discussion would benefit from exploring the broader clinical implications, such as how the presence of these ARGs might impact antibiotic selection and stewardship in dental practice. This could include discussing the risks of horizontal gene transfer (HGT) and the potential spread of resistance beyond the oral cavity.

4. Interpretation of Differential Responses in Microbial Communities: The observed distinct responses among microbial communities in periodontitis versus healthy samples provide important insights. The authors should discuss possible ecological factors (such as changes in nutrient availability or microbial interactions) that could drive these differential responses, and how these factors may guide personalized treatment strategies.

5. Considerations for Antibiotic Guidelines in Periodontal Therapy: The call for guidelines on antibiotic use is well-founded, but it would be valuable for the authors to outline preliminary recommendations based on their findings. For instance, specific guidelines on when to consider alternatives to Amoxicillin, given the high prevalence of resistant Gram-negative anaerobes, would be useful. Including these preliminary recommendations could provide direct, actionable insights for dental professionals.

6. Highlighting Study Limitations and Future Directions: The authors acknowledge the limitations regarding sample size and lack of mutant strain analysis. To build on this, this reviewer suggests recommending specific future studies—such as larger-scale investigations or in vivo studies on interspecies interactions—that could validate and expand upon their findings. Addressing the potential need for longitudinal studies to observe resistance dynamics over time would further enhance the robustness of their recommendations.

7. Detailing Microbial Interactions and Antibiotic Resistance in Polymicrobial Settings: The mention of ecological interactions influencing resistance is valuable, but the authors should further explore how these interactions impact therapeutic outcomes. For instance, discussing specific microbial species interactions and their potential impact on resistance phenotypes in the context of periodontitis could provide new insights into therapeutic planning in biofilm-associated infections.

8. Addressing Clinical Applicability of Metronidazole and Clindamycin Findings: The discussion on Metronidazole’s selective resistance and Clindamycin as a viable alternative to Amoxicillin is insightful. However, it would benefit from a detailed comparison of their clinical applications, including potential side effects, to better guide clinicians in choosing adjunctive therapies for periodontitis.

Conclusions

1. Expand on Key Findings and Clinical Relevance: The conclusion effectively summarizes the role of Streptococcus in tetracycline resistance within periodontitis. Expanding on how this information can impact clinical choices.

2. Highlight Potential for Personalized Treatment Strategies: The conclusion mentions the study’s relevance for personalized antibiotic therapy but does not fully explore the potential applications of these findings. Strengthening this point by suggesting how specific resistance patterns could guide individualized treatment plans or antibiotic selection protocols for severe periodontitis patients would enhance its applicability.

3. Acknowledge Limitations and Implications for Future Research: Adding a brief statement about the study’s limitations, such as sample size or the need for in vivo validation, would balance the conclusions and provide a realistic perspective. Additionally, suggesting specific avenues for future research—like longitudinal studies to track resistance dynamics or exploring other biofilm-associated resistance mechanisms—could demonstrate how this work fits within broader efforts to improve periodontal disease management.

4. Clarify Findings on Non-Periodontal Pathogens: The study’s insight into antibiotic susceptibility among non-periodontal pathogens in periodontitis is valuable but somewhat ambiguous in the current wording. It would be helpful to clarify which non-periodontal pathogens were of concern and how these findings could impact antibiotic prescribing practices, particularly for patients at risk of systemic complications from oral pathogens.

5. Propose Broader Implications for Antibiotic Stewardship in Dentistry: Given the study’s findings, a recommendation to integrate these insights into antibiotic stewardship practices in dentistry would be impactful. This could include a call for more judicious use of antibiotics and monitoring for resistance in periodontal treatment protocols to minimize the spread of resistance.

Reviewer 2 ·

Basic reporting

Very interesting topic that investigated the role of Streptococcus in dental biofilms in exacerbation of periodontitis-specific antibiotic tolerance. In general, the manuscript was well-written in all described sections. However, it needs further proof-reading with grammatical corrections (some of these wrong phrases are highlighted in the attached file. The whole structure, figures and tabled are well- organized and clarified with sharing of the row data.

Experimental design

The current research being within the aims and scope of the journal. The authors were clearly stated the gab of knowledge although it is not fully covered and experimented. However, the methodological analysis that were conduced to achieve the proposed aims were successfully conduced and tried to cover most of the research sections. Nevertheless, some of the methodological sections need to be carefully addressed and re-written as suggested in the attached file.

Validity of the findings

The novelty was fully assessed but not necessarily fully answered as some parts of this work were explain certain criteria of antibiotic resistance but didn't give a definitive answer. This may attributed to complexity of the condition. In spite of this, all the related data were provided in a good way; statistically sound, & controlled.
The conclusions are well-stated, tried to be linked to their results but didn't completely linked and answered the research question because the possibilities of other scenarios can not be excluded

Additional comments

All the comments made to this manuscripts are listed in the original PDF copy which need to be addressed

Annotated reviews are not available for download in order to protect the identity of reviewers who chose to remain anonymous.

---

## Round 0.2 · accepted · Accept

Both reviewers were satisfied with the revisions made to the manuscript.

Reviewer 1 ·

Basic reporting

The authors adequately addressed the requested recommendations.

Experimental design

The authors adequately addressed the requested recommendations.

Validity of the findings

The authors adequately addressed the requested recommendations.

Additional comments

The authors adequately addressed the requested recommendations.

Reviewer 2 ·

Basic reporting

No comments. All previous issues are well addressed by the authors.

Experimental design

No comments. All previous issues are well addressed by the authors.

Validity of the findings

No comments. All previous issues are well addressed by the authors.

Additional comments

No comments. All previous issues are well addressed by the authors.